# A Global Assessment of Sustainable Development Based on Modification of the Human Development Index via the Entropy Method

**Hui Jin [1]** [ORCID]**, Xinyi Qian [2], Tachia Chin [3]** [ORCID] **and Hejie Zhang [1],***

[1] School of Economics, Zhejiang University of Technology, Hangzhou 310023, China; 2111404008@zjut.edu.cn
[2] Qianjiang College, Hangzhou Normal University, Hangzhou 310036, China; 1111704021@zjut.edu.cn
[3] School of Management, Zhejiang University of Technology, Hangzhou 310023, China; tachiachin@zjut.edu.cn
**\*** Correspondence: hzzhj@zjut.edu.cn

**Abstract:** In response to the UN 2030 Agenda for Sustainable Development, this paper proposes a new National Sustainable Development Index (NSDI), based on the modification of the Human Development Index (HDI). The purpose of our research was to improve the widely adopted HDI index by incorporating more comprehensive sustainability perspectives, so as to help policy makers to better analyze the sustainability-related issues facing their countries. After clarifying the concept of sustainable development, our research suggests that this term represents a coordination and configuration of economic, social, and environmental aspects of development, with its major focuses on balancing intra-generational welfare and maximizing the total welfare across generations. We then put forward a novel NSDI framework including 12 indicators from dimensions of economy, resource environment, and society, and calculated the weights of 12 indicators using the entropy method. To further validate our proposed index, this paper also measured the NSDIs of 163 countries in the world, and compared this index with the HDI and other well-known modification indices of HDI. The results showed that the NSDI is a reliable and relative complete index for sustainable development assessment, which makes up for the shortcomings of existing indices.

**Keywords:** sustainable development; environmental assessment; indicators; entropy method

---

## 1. Introduction

With the rapid global development of economy and society, some environmental and social problems, such as the excessive consumption of natural resources, the deterioration of ecological environment, and the imbalance of social development worldwide, have become increasingly serious. To meet these global challenges, the UN 2030 Agenda, which represents an ambitious international step towards sustainable development, was unanimously adopted by all 193 member states [1]. The 2030 Agenda includes 17 Sustainable Development Goals and 169 targets, such as decreasing the global mortality rate of children, lifting more people out of extreme poverty, etc. These Sustainable Development Goals need to be attained via governments' decision-making processes, including policies, plans, programs, and projects. This can be supported by the assessment of sustainable development, understood as a systematic and comprehensive approach that aims to assess the environmental, social, and economic consequences of decision-making. In short, it is necessary to build a relatively systematic and complete composite index with environmental, social, and economic dimensions for sustainability assessment and national development decision-making.

At present, a large number of researchers are devoted to studying the assessment of sustainability or development from different perspectives. Cobb [2] constructed the Index of Sustainable Economic

Welfare (ISEW), which includes environmental and social dimensions, and subsequently made some modifications [3]. Wackernagel and Rees [4] presented the ecological footprint (EF), which is calculated using the ratio of required resources to available resources to measure ecological sustainability. Hamilton et al. [5] developed the genuine savings (GS) concept; it defines the level of re-investment from resource rents that must be reinvested to assure that the (societal) capital stock will never decline. Esty et al. [6] constructed the Environmental Sustainability Index (ESI), which consists of five components and 21 indicators. Esty et al. [7] then amended the ESI by adding indicators of human health and natural resource management, thus creating the Environmental Performance Index (EPI). In addition, many composite indices are constructed by international organizations and used to measure the level of sustainable development by many researchers and governments, such as the the UN's Sustainable Development Goals Index [8], the UNDP's (United Nations Development Programme, a department of a United Nations Organization) Human Development Index [9], and so on. Recently, some researchers have begun to pay attention to sustainable development at the regional level (e.g., [10–14]) . Nevertheless, it is still very important to reveal and measure the actual status of sustainable development at the national level for national development policy-making.

The Human Development Index (HDI) is the one of the most widely used and referenced indices for assessing sustainable development and ranking different countries [13,15]). The HDI is an excellent composite index, and famous for its simple composition, representative sub-indicators, and rich connotation. It consists of three (equally weighted) sub-indicators: income, life expectancy, and education [9]. However, there is a serious problem if the HDI used as a sustainable development index—that is, a lack of environmental indicators [16]. Thus, some indices have amended this issue by adding indicators of environmental and ecological aspects, such as the Human Sustainable Development Index [17] (Bravo, 2014) and the Human Green Development Index [18] (Li et al., 2014). The Human Sustainable Development Index (HSDI) adds per capita $CO_2$ emissions as an environmental indicator to the HDI. The Human Green Development Index (HGDI) includes 12 indicators of social–economic and resource–environment dimensions, of which 6 indicators represent resource and environmental aspects, namely $CO_2$ emissions, $PM_{10}$ (particles less than 10 microns in diameter), proportion of forest area, proportion of threatened animals, proportion of land conservation area, and utilization ratio of primary energy [18]. Another six indicators in the HGDI present social and economic dimensions, namely the proportion of the population below the minimum food energy intake standard, the income index, life expectancy, the education index, the population's access to improved health facilities and the population's access to improved drinking water. HGDI and HSDI are the two of the most recent and widely cited modifications of HDI.

However, HSDI and HGDI, as improvements of HDI, are still incomplete, and have shortcomings to varying degrees. Firstly, the HSDI and HGDI are not complete enough, because they include only income index as an economic indicator, which is not strong enough to capture a complex economic system. Sustainable development, to a great extent, not only protects the environment, but also serves to improve social cohesion and economic growth [1]. As *Our Common Future* defines, sustainable development is to balance the welfares between present generations and future generations (World Commission on Environment and Development, 1987). Therefore, we should pursue economic growth to ensure the welfare of present generations, while protecting the ecological environment and rationally utilizing natural resources to ensure the welfare of future generations. If were to only protect the environment and make the economy stagnate, this would also not represent a sustainable development mode. Secondly, the HSDI still raises doubts about its capacity to capture the complexity of a coupled human–environment system, because it only adds the per capita $CO_2$ emissions as an environmental indicator [13]. Thirdly, the 12 indicators of HGDI are equally weighted, which is not objective enough, as its creators have pointed out [18].

Therefore, this paper aimed to propose a relatively systematic and complete index of sustainable development, that is, the National Sustainable Development Index (NSDI), to make up for the gaps of existing indices and to help governments make better national development decisions. Firstly,

our research will be helpful to strengthen the planning and understanding of sustainable development. Many studies have paid attention to the environmental dimension of sustainable development while ignoring the essence of sustainable development, namely comprehensive and coordinated development in economic, environmental, and social dimensions [1]. Secondly, this paper built the NSDI using the Entropy method to make the calculation of weights more scientific and objective. Thirdly, the NSDI makes sense as a more complete index of sustainable development by strengthening social, environmental, and economic dimensions, and thus represents a small step ahead of the HDI and other existing indices.

The rest of the paper is organized into five sections. Section 2 redefines the concept of sustainable development. Section 3 describes the construction of NSDI and the introduction of the entropy method, which was used to calculate the weights of the indicators. Section 4 describes the measurement of NSDI. Section 5 displays the comparison of the NSDI with other indices. Section 6 is the conclusion.

## 2. Sustainable Development

The concept of sustainable development originated from ecology, although it has recently brought together many disciplines and interests, involving ecology together with environmental, economic, and societal aspects [14,19,20]. Thus, it is necessary to review the relevant research on sustainable development and redefine the concept of sustainable development.

In research into sustainable development, researchers in different disciplines have different perspectives and emphases [21]. Ecologists and environmentalists study sustainable development from the aspects of ecological environment pollution, biodiversity, and ecosystem optimization, and they focus on the long-term and healthy survival of human beings as well as the sustainability of ecosystems and the regional environment (e.g., [10,11,22–25]). Economists reveal the root causes of population, poverty, environment, energy, and growth problems, and use economic theories and methods to explore how to activate economic power to promote sustainable development—Ranis et al. [26], Bilbao-Ubillos [27], Bolcárová and Kološta [14], Zhang et al. [28] worked from this perspective. Sociologists, like Ma et al. [29], Bergman at al. [1], emphasize how to establish a structural system including market, policy, moral standards, science and technology, and other factors, which could maximize the cohesion of nature, humanity, and society to the track of sustainable development.

Therefore, researchers of different disciplines define sustainable development differently [20]. Ye and Luan [30] pointed out that sustainable development means to continuously improve the quality of human life and environmental bearing capacity, simultaneously meet the needs of present generations and that of future generations, and to meet the needs of people from different regions and countries. They believed that the core element of sustainable development is fairness (or balance), which should be both intra-generational and contemporary. Fang et al. [31] and Zhang [32] also came to a similar definition with Ye and Luan [30]. Chen [33] and Zeng et al. [34] argued that sustainable development includes three basic elements, namely economy, ecology, and society. In addition, they suggested that we should set ecological development as the premise, economic development as the method, and social development as the purpose, to make human society and the ecosystem develop harmoniously. Sociologists and ecologists emphasize "harmony" and "fairness", while economists prefer to use economic concepts like growth, utility, and welfare to represent sustainable development. According to Dasgupta [35], the maximization of inter-generational utility is equivalent to balancing the welfare between present and future generations; thus, sustainable development maximizes the total utility of all generations. Peng and Bao [36] pointed out that sustainable development is a way of development in which the per capita welfare increases, or at least does not decrease over time. Lin and Yang [37] defined sustainable development as maximizing economic welfare under the conditions of ecological protection and rational utilization of resources.

Although the definitions of sustainable development are different, the essence is the same; sustainable development is to coordinate economic, social, and environment development, to balance

the intra-generational welfare, and to maximize the total welfare of generations. Furthermore, there are two issues to which attention should still be paid.

The first issue is the question of what the principle of inter-generational equity (or balance) is, and how to reflect it in the construction of a sustainable development index. Economists have explained it through the utility theory. As Dasgupta [35] has pointed, the maximization of inter-generational utility is equivalent to balancing the welfare between present and future generations; thus, sustainable development is to maximize the total utility of all generations. This indicates that inter-generational equity means the maximization of total utility of all generations. Obviously, this explanation is so theoretical and abstract that it could not be used as a practical principle in the construction of sustainable development index, but its connotation is easy to understand and accept. First, we should pursue economic growth and social progress to ensure the welfare of present generations, while protecting the ecological environment and rationally utilizing natural resources to ensure the welfare of future generations. Furthermore, the weight of economic and social dimensions should be close to that of resource and environmental dimensions. Thus, inter-generational equity could be reflected in both the connotation and weights of a sustainable development index [18].

The second issue is how to choose indicators to measure the sustainability of the three dimensions. In order to address this problem, we selected some representative composite indices which are widely referenced and used for sustainability assessment, such as the UNDP's HDI, the UN's SDG index, the HSDI, the HGDI, and many other indices published in mainstream journals. We studied and summarized the factors or indicators in these indices, hoping to collect some experience and rules from them. As Table 1 shows, income, employment, economic structure, and economic growth are common indicators used to measure sustainable development in the economic dimension, while land protection, energy consumption, $CO_2$ emissions, water protection, and air quality are always used to represent sustainability in resource and environmental dimension. In the social dimension, education, health, potable water, poverty, and sanitation are practical and important indicators. These existing studies provided us with references and guidance, and helped us find some ideas to address the problem. However, we still needed to deeply understand the concept and connotation of sustainable development and the 17 Sustainable Development Goals, and to build a set of scientific indicator selection criteria.

**Table 1.** The factors or sub-indicators in some existing composite indices.

| Authors | Factors or Indicators in Composite Indices | | |
| --- | --- | --- | --- |
| | Economic Dimension | Resource and Environmental Dimension | Social Dimension |
| Adrián and Américo (2002) | Gross Domestic Product (GDP), employment | Air quality, land use, protected areas | Education, health, poverty, potable water, sewage infrastructure |
| UNDP (2004) | Income | | Education, health |
| Kondyli (2010) | Economic structure, size of economy | Potable water, sea quality, land quality, biodiversity | Population size, population structure, poverty, unemployment |
| Li et al. (2014) | Income | Primary energy consumption, $CO_2$ emissions, $PM_{10}$ (particles less than 10 microns in diameter), forest, threatened animals, land conservation | Education, health, potable water, sanitation facilities, poverty |
| Bravo (2014) | Income | $CO_2$ emissions | Education, health |
| Bolcárová andOlošta (2015) | Economic growth | Resource productivity, greenhouse gas emissions, renewable energy consumption, natural resources | Social inclusion, health |

**Table 1.** *Cont.*

| Authors | Factors or Indicators in Composite Indices | | |
| --- | --- | --- | --- |
| | **Economic Dimension** | **Resource and Environmental Dimension** | **Social Dimension** |
| UN (2015) | Economic growth and employment, infrastructure, city and communities | Energy consumption, $PM_{2.5}$ (particles less than 2.5 microns in diameter), $CO_2$ emissions, marine protection, land protection | Poverty, trophic level, health, education, gender equality, water and sanitation, inequality, peace and justice |
| Guo et al. (2016) | Income, economic growth, economic structure, Foreign Direct Investment (FDI), public revenue | Arable land, water quality, primary energy consumption, land conservation, air quality | Living standard, education, social security, safety |

Note: (1) For some composite indices with too many sub-indicators, we have presented only their factors or important and symbolic sub-indicators. Additionally, some sub-indicators are named differently in different indices, such as "drinking water" and "potable water". In order to effectively summarize the above indices, we renamed some factors or indicators, but did not change their connotation. (2) Guo, C.Z.; Peng, Z.Y.; Ding, J.Q. Construction of the Indexes of DEA Used in Comprehensive Evaluation of Sustainable Development. *China Popul. Resour. Environ.* **2016**, *3*, 9–17.

## 3. The Construction of NSDI

### 3.1. Criteria for Choosing Indicators

This paper aimed to build a concise and acceptable composite index that reflects the triple sustainable development dimensions of economy, society, and environment. Therefore, we selected a battery of relevant indicators in accordance with the following criteria based on the wide range of absorption at the experience of predecessors.

1. The indicators of sustainable development index should include economic, resource, environmental, and social dimensions [38].
2. The selected indicators should be representative [23,39]. It is better to choose existing indicators.
3. The quantity of indicators should not be too many, making the NSDI concise and acceptable [18].
4. Indicators should be continuous and comparable to make the NSDI could be comparable by country and time [40,41].
5. The selected indicators must be quantifiable and have strong operability [42].
6. Availability and reliability of the source of data [10].

### 3.2. The Framework of NSDI

According to the concept of sustainable development, the Sustainable Development Goals and the existing studies, this paper established the NSDI with three dimensions, namely economy, society, and resource and environment.

#### 3.2.1. Economic Dimension of the NSDI

Governments should pursue a relatively high and fair income for folks, a potential for economic growth and a reasonable economic structure to improve the welfare of the present generation [43]. Accordingly, indicators of income level, economic growth, and economic structure need to be set. On one hand, income indicates the current level of economic development. Of course, the current level will affect further development in the future. On the other hand, economic growth and economic structure represent the potential for future economic development. The two indicators reflect the competitiveness of its economic activities [11]. This competitiveness shapes an economic base that is supported by dynamic local activities.

### 3.2.2. Social Dimension of the NSDI

The government should not only improve social welfare, but should also consider social fairness and harmony. Thus, education for the young, medical treatment for the sick, basic sanitation, and drinking water should be guaranteed. The education and healthcare provided by governments represent social welfare for all residents. In particular, for residents in poor families, education is an important channel for their future development, while healthcare is the basic guarantee for their life and health. Additionally, basic sanitation and drinking water are the most basic requirements for human survival. Therefore, the above four factors not only reflect social welfare, but also represent the consideration of government for social fairness and harmony.

### 3.2.3. Resource and Environmental Dimension of the NSDI

The resource and environment dimension, through the services it provides to society and the economy, has an effect on the performance of economic activities and on the psychosomatic condition of people [11]. Moreover, this dimension of sustainability also reflects the welfare guarantee for future generations. Hence, the protection of the environment and the utilization of resources are important and are associated with the preservation of their quantitative and qualitative characteristics. The climate and air quality not only reflect the living conditions and quality of human beings in the present generation, but also affect that of future generations, while the forest, arable land, and energy consumption represent the current resource and environmental conditions, and affect the performance of economic activities. Furthermore, these five factors also reflect the insurance of welfare for future generations.

Finally, we built the framework of NSDI to include three dimensions and 12 factors (see columns 1 to 3 in Table 2). According to Li et al. [18], the construction of a sustainable development index should embody the idea of humanistic care and a people-oriented development mode rather than the pursuit of scientism.

**Table 2.** The Sustainable Development Evaluation Index.

| Index | Dimension | Factor | Indicator | Premise |
|---|---|---|---|---|
| | Economic dimension | Economic growth | Real GDP growth | + |
| | | Income level | Income index | + |
| | | Economic structure | Employment in services (% of total employment) | + |
| National Sustainable Development Index (NSDI) | Resource and environmental dimension | Climate | CO$_2$ emissions per capita | - |
| | | Air quality | PM2.5 | - |
| | | Forest | Forest area (% of total land area) | + |
| | | Arable land | Arable land per person | + |
| | | Energy | Renewable energy consumption (% of total final energy consumption) | + |
| | Social dimension | Education | Mean years of schooling | + |
| | | Health | Life expectancy index | + |
| | | Drinking-water | Population using improved drinking-water sources (%) | + |
| | | Sanitation | Population using improved sanitation facilities (%) | + |

Note: detailed descriptions of the 12 indicators can be found in Table 3.

### 3.3. The Selection of the 12 Indicators in NSDI

The representativeness and typicality of the selected indicators (variables) are related to the measurement and practical value of the NSDI. Thus, it was very important to choose one indicator in each of the 12 areas related to sustainable development. According to the criteria for choosing indicators, and referring to the advanced practices of well-known indices, we formulated meticulous operation steps for indicator selection. Taking the selection process of the "Education" indicator as an example, the details are as follows.

### 3.3.1. Searching Relative Indicators for Each Factor

There are more than 20 indicators for the factor of "Education", such as "Government expenditure on education", "Government expenditure per student", "Gross intake ratio in first grade of primary education", "Literacy rate (adult)", "Progression to secondary school", "School enrolment, secondary", "primary School enrolment", "Trained teachers in primary education", "Primary completion rate", "Mean years of schooling", and so on. We studied and compared these indicators, and chose the most representative and suitable indicator in each field based on the selection criteria and existing well-known indices.

### 3.3.2. Comparing All Indicators

According to the indicator selection criteria above, we compared all indicators and investigated their representativeness, comparability, continuity, and availability. For example, "Government expenditure on education" can represent government spending and emphasis on education, but cannot effectively measure current education quality and future education development. The data for "Trained teachers in primary education" are not available in more than 120 sovereign states. Fortunately, these indicators for education are all continuous and comparable. Thus, we eliminated the indicators that lacked of representativeness and availability.

### 3.3.3. Choosing the Most Suitable and Representative Indicator

Due to the third criterion, we chose only one indicator for education in order to make the NSDI concise and easily accepted; thus, that indicator had to be the most suitable and representative one. "Literacy rate (adult)" and "Mean years of schooling" are relatively representative and available as education indicators, and they are widely used to measure the education level of a country. We finally chose the "Mean years of schooling" as the education indicator. The first reason is that adult literacy rate is not "fair" for developing countries, and could not represent future education development. Many developing countries became independent after World War II, some even in the 1990s. The older generation in these countries grew up in chaotic wartime, which led to a very low literacy rate. Although the "Mean years of schooling" will be affected by the age structure too, as an average indicator, the impact of age structure on it can be minimized to a large extent. Secondly, adult literacy rate lacks differentiation, especially for countries with a high economic development level, where the level reaches almost 100%. Thirdly, we were able to gather more samples if we chose the "Mean years of schooling" indicator.

The selection process of the "Education" indicator is briefly described above. It was similar to the selection process of the remaining 11 indicators. Due to the limitation of space, we will not explain the selection process of each indicator in detail. As Table 2 shows, the NSDI is a simple and clear systematic composite index with three dimensions and 12 indicators. These 12 indicators, selected from 12 aspects, justify the importance of sustainable development in the economic, environmental, and social dimensions. They are the most basic and primary goals for human economic and social development, for the protection of the world's environment and for sustainable utilization of natural resources. The meaning, units, and data sources of the 12 indicators are presented in Table 3.

**Table 3.** Introduction and Data Sources of the 12 Indicators.

| Indicators | Meaning and Data Source |
|---|---|
| GDP Growth | Real GDP growth.<br>Data source: World Bank database (https://data.worldbank.org/indicator) |
| Income index | According to Atkinson [44], calculating the income index can reflect fairness and equality in the case of unequal distribution factors, based on the disposable income or consumption of per capita family. The higher the income index is, the better the economic situation of the country is, and the more equal and fairer the income distribution of the country is.<br>Data source: The UNDP database (http://hdr.undp.org/en/data#) |
| Employment in services (% of total employment) | The proportion of employment of the tertiary industry in total employments, which is used to measure the economic structure.<br>Data source: The UNDP database (http://hdr.undp.org/en/data#) |
| Per capita $CO_2$ emissions | $CO_2$ emissions generated by the combustion of energy such as coal, oil, natural gas, and so on (unit: ton per person).<br>Data source: International Energy Agency (http://www.iea.org/) |
| $PM_{2.5}$ | The concentration in the atmosphere of fine suspended particles with a diameter less than 2.5 microns, which can penetrate into the respiratory tract and cause serious health damage (unit: microgram/m$^3$).<br>Data source: World Bank database (https://data.worldbank.org/indicator) |
| Forest coverage rate | The forest coverage rate is the proportion of forest area in the total land area, while the forest area refers to the land covered by upright trees (at least 5 m) which grow naturally or are planted artificially.<br>Data source: The UNDP database (http://hdr.undp.org/en/data#) |
| Arable land per person | Arable land includes temporary crop land (double-cropping rice field is calculated once), temporary grassland for mowing or pasture, market or kitchen garden land, and temporary fallow land, but excludes land abandoned due to rotation.<br>Data source: World Bank database (https://data.worldbank.org/indicator) |
| Renewable energy consumption | The proportion of renewable energy consumption in total energy consumption. The higher the proportion is, the more conducive to the sustainable development in resources and environmental dimension.<br>Data source: The UNDP database (http://hdr.undp.org/en/data#) |
| Mean years of schooling | Mean years of education for adults over 25 years old (unit: years).<br>Data source: The UNDP database (http://hdr.undp.org/en/data#) |
| Life expectancy index | According to Atkinson [44], calculating the life expectancy index can reflect fairness and equality in the case of unequal distribution factors, based on the data of UN life table. The higher the index value, the better the health status of residents, and the more equal and fairer the access to healthcare for residents.<br>Data source: The UNDP database (http://hdr.undp.org/en/data#) |
| Population using improved drinking water sources (%) | An improved drinking water source is a drinking water source that is free from external pollution, especially from excreta pollution, due to its own structure or through active intervention.<br>Data source: The World Health Organization (http://www.wssinfo.org/data-estimates/table/) |
| Population using improved sanitation facilities (%) | The proportion of the population with basic excreta treatment facilities, which can effectively prevent human, livestock, mosquitoes, and flies from contacting with excreta. Improved sanitation facilities include simple but protected latrines, and direct flush latrines connected to sewer lines, of which normal function can be guaranteed.<br>Data source: The World Health Organization (http://www.wssinfo.org/data-estimates/table/) |

### 3.4. Entropy Method for Calculating the Weights of Indicators

The entropy method is a method used to calculate the weight of each indicator in a composite indicators system, based on the idea of entropy from basic information theory. Specifically, information is a measure of the degree of order in a system and entropy is a measure of the degree of disorder in a system; therefore, the smaller the indicator information entropy, the greater the information provided by the indicator, the greater its effect in the comprehensive evaluation, and the higher the weight [45,46]. According to Zhang et al. [45], the weight calculated by the entropy method represents the relative rate of change of the indicator in a composite indicators system, while the relative level of each indicator should be figured by the standardized value of its data. Thus, the entropy method is an objective weighting method that makes weight judgments based on the size of the data information load. It can reduce the influence of human subjectivity on the evaluation result and makes the evaluation results more realistic [29,46]. Therefore, this method could make up for the lack of objectivity that HDI and HGDI have due to their use of the subjective evaluation method to calculate the weights of indicators.

According to the introduction of the entropy method above, we needed to relate the different variables in different units with a dimensionless scale from 0 to 1. As shown in Equation (1), $x_{ij}$ is the indicator $j$ of country $i$, and $\widetilde{x}_{ij}$ is the result of dimensionless treatment. It should be noted that for some indicators, like per capita $CO_2$ emissions, higher values mean a poorer performance of sustainable development, which need to be treated using Equation (2).

$$\widetilde{x}_{ij} = \frac{X_{ij} - minX_{ij}}{maxX_{ij} - minX_{ij}} \tag{1}$$

$$\widetilde{x}_{ij} = 1 - \frac{X_{ij} - minX_{ij}}{maxX_{ij} - minX_{ij}} \tag{2}$$

Secondly, the entropy value of each indicator is calculated, as shown in Equations (3) and (4). $e_j$ is the entropy value of each indicator.

$$k = 1 / \ln(n) \tag{3}$$

$$e_j = -k \sum_{i=1}^{n} \widetilde{x}_{ij} ln\widetilde{x}_{ij} \tag{4}$$

Thirdly, the information utility value of each indicator, namely $g_j$, is calculated.

$$g_j = 1 - e_j \tag{5}$$

Finally, the weight of indicator $j$ is obtained, namely $\omega_j$, as shown in Equation (6).

$$\omega_j = g_j / \sum_{j=1}^{p} g_j \tag{6}$$

## 4. The Measurement of NSDI

We chose to measure the NSDI of 163 countries in 2015. These countries were chosen according to two criteria: (1) all countries had published the data of all 12 indicators; (2) internationally recognized non-sovereign entities were not selected, such as Taiwan, China. Additionally, we chose to measure the NSDI with the latest data as of 2015, such as the data of "Population using improved drinking-water sources (%)" and "Population using improved sanitation facilities (%)". In general, the 163 selected countries included most of the sovereign countries in the world, so this study has statistical significance.

According to the framework of NSDI, this paper measured the NSDI and its ranking of 163 countries in 2015 with the entropy method through Stata 15.0. As the weights in Table 4 show, the economic dimension, social dimension, and resource–environmental dimension respectively

accounted for 24.60%, 23.93%, and 51.46% (resource dimension was 22.87%, environmental dimension was 28.59%). According to the concept of sustainable development, we should pursue economic growth and social progress to ensure the welfare of present generations, while protecting the ecological environment and rationally utilizing natural resources to ensure the welfare of future generations. If we were to protect the environment and make the economy stagnate, this would also not be a sustainable development mode. Obviously, the results of weight calculation coincided with our theoretical viewpoint. The sum of the weights of the economic and social dimensions was almost equal to the weights of the resource–environmental dimensions. This represents the concept and essence of sustainable development—that the welfare of the present and future generations is equally important, and that we should not "care for this and lose that". Additionally, resource and environment are important factors of economic development and contribute to quality of life, which justifies this high weight.

**Table 4.** The Weights of the 12 Indicators.

| Index | Dimension | Factor | Indicator | Weights |
|---|---|---|---|---|
| National Sustainable Development Index (NSDI) | Economic dimension | Economic growth | Real GDP growth | 6.09% |
| | | Income level | Income index | 9.20% |
| | | Economic structure | Employment in services (% of total employment) | 9.31% |
| | Resource and environmental dimension | Climate | $CO_2$ emissions per capita | 12.30% |
| | | Air quality | $PM_{2.5}$ | 7.55% |
| | | Forest | Forest area (% of total land area) | 8.74% |
| | | Arable land | Arable land per person | 14.49% |
| | | Energy | Renewable energy consumption (% of total final energy consumption) | 8.38% |
| | Social dimension | Education | Mean years of schooling | 7.14% |
| | | Health | Life expectancy index | 7.39% |
| | | Drinking water | Population using improved drinking water sources (%) | 4.95% |
| | | Sanitation facilities | Population using improved sanitation facilities (%) | 4.45% |

As the result in Table 5 shows, the top five countries in the NSDI were Australia (0.747), Norway (0.746), Switzerland (0.736), Denmark (0.729), and Canada (0.693), while the bottom five countries were Bahrain (0.342), Kuwait (0.326), Mozambique (0.305), Niger (0.260), and Cote d'Ivoire (0.232). The average was 0.545 and the index distribution was significantly left-skewed (−0.108).

The NSDI of each country showed distinct characteristics in economic level and geographical distribution. Countries with high NSDI tended to be developed countries, which are mainly in Europe, North America, and Oceania. In the NSDI ranking, 14 of the top 25 countries were European developed countries. By contrast, the last 25 countries were developing countries mainly in Asia and Africa. This means that the sustainable development level of developing countries was generally lower than that of developed countries. There are three main reasons for the low level of sustainable development in developing countries. The first is the low level of the economy and residents' income. The second is the insufficient supply of public goods, such as education, medical care, public hygiene, environmental protection, etc. Last but not least, some developing countries, such as China, are bombarded with such problems as inadequate management and technology of pollution control and resource utilization, while still promoting economic growth at all costs, which leads to serious damage to resources and environment [47].

Furthermore, the NSDI of emerging market countries was generally low. Emerging market countries refer to those countries with gradually improving market economic systems, high economic growth rate, and great potential. Although these countries are developing rapidly and playing an increasingly important role in the international community, their sustainable development level is still relatively low, for that their governments pay more attention to GDP than to people's livelihoods, natural resources, or the environment.

Similarly, the NSDI was also relatively low in rich Middle East countries. These countries have very high economic levels and residents' income levels because of the huge oil resources. Nevertheless,

their performance in NSDI rankings was very bad, such as Qatar (141), United Arab Emirates (149), and Saudi Arabia (156), due to overdependence on and consumption of natural resources.

**Table 5.** The NSDI and Ranking of the 163 Countries in 2015.

| Country | NSDI | Rank | C | DC | EC | Country | NSDI | Rank | C | DC | EC |
|---|---|---|---|---|---|---|---|---|---|---|---|
| Australia | 0.747 | 1 | OC | Y | N | Armenia | 0.543 | 83 | AS | N | N |
| Norway | 0.746 | 2 | EU | Y | N | Kyrgyzstan | 0.543 | 84 | AS | N | N |
| Switzerland | 0.736 | 3 | EU | N | N | Sao Tome and Principe | 0.542 | 85 | AF | N | N |
| Denmark | 0.729 | 4 | EU | Y | N | El Salvador | 0.542 | 86 | NA | N | N |
| Canada | 0.693 | 5 | NA | Y | N | Gambia | 0.541 | 87 | AF | N | N |
| Sweden | 0.690 | 6 | EU | Y | N | Vanuatu | 0.539 | 88 | OC | N | N |
| Latvia | 0.687 | 7 | EU | N | N | Papua New Guinea | 0.535 | 89 | AF | N | N |
| Japan | 0.683 | 8 | AS | Y | N | Congo | 0.535 | 90 | AF | N | N |
| United States | 0.681 | 9 | NA | Y | N | Malawi | 0.534 | 91 | AF | N | N |
| Germany | 0.679 | 10 | EU | Y | N | Barbados | 0.532 | 92 | NA | N | N |
| Serbia | 0.679 | 11 | EU | N | N | Azerbaijan | 0.527 | 93 | AS | N | N |
| Italy | 0.677 | 12 | EU | N | N | Tunisia | 0.524 | 94 | AF | N | N |
| Finland | 0.675 | 13 | EU | Y | N | Timor-Leste | 0.524 | 95 | AS | N | N |
| New Zealand | 0.674 | 14 | OC | Y | N | Botswana | 0.523 | 96 | AF | N | N |
| Lithuania | 0.673 | 15 | EU | N | N | Samoa | 0.522 | 97 | OC | N | N |
| France | 0.671 | 16 | EU | Y | N | Namibia | 0.522 | 98 | AF | N | N |
| United Kingdom | 0.665 | 17 | EU | Y | N | Dominican Republic | 0.520 | 99 | NA | N | N |
| Kazakhstan | 0.663 | 18 | AS | N | N | Iran | 0.519 | 100 | AS | N | N |
| Luxembourg | 0.661 | 19 | EU | Y | N | Maldives | 0.517 | 101 | AS | N | N |
| Ireland | 0.660 | 20 | EU | Y | N | Ghana | 0.516 | 102 | AF | N | N |
| Belgium | 0.655 | 21 | EU | Y | N | Suriname | 0.515 | 103 | SA | N | N |
| Portugal | 0.655 | 22 | EU | Y | N | Lebanon | 0.514 | 104 | AS | N | N |
| Iceland | 0.651 | 23 | EU | N | N | Morocco | 0.513 | 105 | AF | N | Y |
| Netherlands | 0.649 | 24 | EU | Y | N | Cameroon | 0.512 | 106 | AF | N | N |
| Korea (Rep.) | 0.647 | 25 | AS | Y | Y | Tajikistan | 0.511 | 107 | AS | N | N |
| Argentina | 0.647 | 26 | SA | N | N | Jordan | 0.510 | 108 | AS | N | N |
| Malta | 0.641 | 27 | EU | N | N | Rwanda | 0.510 | 109 | AF | N | N |
| Spain | 0.641 | 28 | EU | Y | N | Haiti | 0.509 | 110 | NA | N | N |
| Israel | 0.633 | 29 | AS | N | N | Senegal | 0.509 | 111 | AF | N | N |
| Singapore | 0.631 | 30 | AS | Y | N | Kenya | 0.509 | 112 | AF | N | N |
| Brazil | 0.629 | 31 | SA | N | Y | Peru | 0.507 | 113 | SA | N | Y |
| Belize | 0.621 | 32 | NA | N | N | Angola | 0.505 | 114 | AF | N | N |
| Montenegro | 0.615 | 33 | EU | N | N | Togo | 0.503 | 115 | AF | N | N |
| Fiji | 0.615 | 34 | OC | N | N | Eswatini | 0.501 | 116 | AF | N | N |
| Austria | 0.614 | 35 | EU | Y | N | Benin | 0.500 | 117 | AF | N | N |
| Estonia | 0.614 | 36 | EU | N | N | Cabo Verde | 0.499 | 118 | AF | N | N |
| Greece | 0.613 | 37 | EU | N | N | South Africa | 0.499 | 119 | AF | N | Y |
| Belarus | 0.612 | 38 | EU | N | N | Comoros | 0.498 | 120 | AF | N | N |
| Gabon | 0.612 | 39 | AF | N | N | Mali | 0.497 | 121 | AF | N | N |
| Hungary | 0.610 | 40 | EU | N | Y | Burkina Faso | 0.495 | 122 | AF | N | N |
| Brunei Darussalam | 0.608 | 41 | AS | N | N | Turkmenistan | 0.495 | 123 | AS | N | N |
| Romania | 0.608 | 42 | EU | N | N | Nigeria | 0.493 | 124 | AF | N | N |
| Bulgaria | 0.607 | 43 | EU | N | N | Oman | 0.489 | 125 | AS | N | N |
| Lao PDR | 0.604 | 44 | AS | N | N | Madagascar | 0.488 | 126 | AF | N | N |
| Croatia | 0.604 | 45 | EU | N | N | Lesotho | 0.485 | 127 | AF | N | N |
| Ukraine | 0.602 | 46 | EU | N | N | Nepal | 0.485 | 128 | AS | N | N |
| Bhutan | 0.602 | 47 | AS | N | N | Pakistan | 0.485 | 129 | AS | N | N |
| Slovenia | 0.600 | 48 | EU | N | N | Moldova | 0.481 | 130 | EU | N | N |
| Algeria | 0.594 | 49 | AF | N | N | China | 0.476 | 131 | AS | N | Y |
| Russian | 0.593 | 50 | EU | N | Y | Uganda | 0.473 | 132 | AF | N | N |
| Slovakia | 0.592 | 51 | EU | N | N | Bangladesh | 0.471 | 133 | AS | N | N |
| Albania | 0.587 | 52 | EU | N | N | Libya | 0.469 | 134 | AF | N | N |
| Bosnia and Herzegovina | 0.585 | 53 | EU | N | N | Equatorial Guinea | 0.458 | 135 | AF | N | N |
| Turkey | 0.585 | 54 | AS | N | Y | Iraq | 0.456 | 136 | AS | N | N |
| Bolivia | 0.583 | 55 | EU | N | N | Solomon Islands | 0.453 | 137 | OC | N | N |
| Colombia | 0.581 | 56 | SA | N | Y | Guyana | 0.453 | 138 | SA | N | N |
| Uruguay | 0.578 | 57 | SA | N | N | Egypt | 0.449 | 139 | AF | N | Y |
| Honduras | 0.577 | 58 | NA | N | N | Mauritania | 0.436 | 140 | AF | N | N |
| Cambodia | 0.576 | 59 | AS | N | N | Qatar | 0.432 | 141 | AS | N | N |
| Georgia | 0.573 | 60 | AS | N | N | Guinea-Bissau | 0.430 | 142 | AF | N | N |

**Table 5.** *Cont.*

| Country | NSDI | Rank | C | DC | EC | Country | NSDI | Rank | C | DC | EC |
|---|---|---|---|---|---|---|---|---|---|---|---|
| Czechia | 0.573 | 61 | EU | N | Y | Afghanistan | 0.427 | 143 | AS | N | N |
| Poland | 0.571 | 62 | EU | N | Y | Guinea | 0.419 | 144 | AF | N | N |
| Guatemala | 0.569 | 63 | NA | N | N | Sierra Leone | 0.415 | 145 | AF | N | N |
| Indonesia | 0.569 | 64 | AS | N | Y | Yemen | 0.414 | 146 | AS | N | N |
| Panama | 0.569 | 65 | NA | N | N | Congo (Dem. Rep.) | 0.413 | 147 | AF | N | N |
| Chile | 0.567 | 66 | SA | N | Y | Zimbabwe | 0.412 | 148 | AF | N | N |
| Mongolia | 0.566 | 67 | AS | N | N | United Arab Emirates | 0.410 | 149 | AS | N | N |
| Cyprus | 0.565 | 68 | EU | N | N | Chad | 0.406 | 150 | AF | N | N |
| Paraguay | 0.564 | 69 | SA | N | N | Ethiopia | 0.401 | 151 | AF | N | N |
| Bahamas | 0.564 | 70 | NA | N | N | Liberia | 0.397 | 152 | AF | N | N |
| Ecuador | 0.563 | 71 | SA | N | N | Central African Republic | 0.392 | 153 | AF | N | N |
| Mexico | 0.563 | 72 | SA | N | Y | Burundi | 0.392 | 154 | AF | N | N |
| Malaysia | 0.561 | 73 | AS | N | Y | Trinidad and Tobago | 0.391 | 155 | NA | N | N |
| India | 0.560 | 74 | AS | N | Y | Saudi Arabia | 0.390 | 156 | AS | N | N |
| Thailand | 0.560 | 75 | AS | N | Y | Zambia | 0.383 | 157 | AF | N | N |
| Jamaica | 0.556 | 76 | NA | N | N | Nicaragua | 0.376 | 158 | NA | N | N |
| Tanzania | 0.553 | 77 | AF | N | N | Bahrain | 0.342 | 159 | AS | N | N |
| Mauritius | 0.552 | 78 | AF | N | N | Kuwait | 0.326 | 160 | AS | N | N |
| Viet Nam | 0.551 | 79 | AS | N | N | Mozambique | 0.305 | 161 | AF | N | N |
| Philippines | 0.550 | 80 | AS | N | Y | Niger | 0.260 | 162 | AF | N | N |
| Tonga | 0.549 | 81 | OC | N | N | Cote d'Ivoire | 0.232 | 163 | AS | N | N |
| Myanmar | 0.548 | 82 | AS | N | N | | | | | | |

Note: (1) C refers to the continent, so AS is Asia, AF is Africa, EU is Europe, NA is North America, SA is South America, OC is Oceania. (2) DC indicates whether it is a developed country, which according to the standards of CIA's *the world Fact Book* and IMF. (3) EC indicates whether it is an emerging market country, referring to the MSCI Emerging Market Index in 2009. (4) Thus, Y is short for yes, and N is short for no.

## 5. Discussion

To assess whether the NSDI could help policymakers and government officials in their decision-making toward achieving an all-round sustainable development goal, we compared it with HDI, HSDI, and HGDI at national level.

Since 1990 the HDI is reported annually as part of the Human Development Report of the UNDP, and has gradually become a widely used and cited index for sustainability assessment due to its simple composition and rich connotation [27]. It consists of three (equal weighted) sub-indices which are aggregated by an arithmetic mean: education, income and life expectancy. Although the composition is simple, its connotation is very rich. The HDI is based on the theory of welfare economics with fairness and substantial freedom, which contains a deep understanding of the main concept of human development [18]. In the past, the traditional meaning of "development" was strictly economic, as it dealt only with the economic side of development [27]. For instance, per capita GDP used to be a basic indicator for development trend and level. In subsequent years, more and more scholars have moved towards a new concept of development in which economic growth is seen as a condition that is necessary but not sufficient to explain the degree of development of a country [48,49]. And they pay more attention to the real welfare that people enjoy, namely human sustainable development. The essential abilities for human development are therefore the abilities to lead a long, healthy life, to obtain knowledge, to access the resources needed for a decent standard of living, and to take part in the life of the community [27]. Based on the above theories and ideas, the HDI is born to measure the human development in national level. Actually, more and more scholars believe that human-oriented development mode is the sustainable development mode. Therefore, the HDI gradually becomes one of the most widely used composite index for measuring sustainability.

HSDI, HGDI and NSDI are regarded as "derivative indices" or modification schemes of the HDI, but they are quite different in composition and connotation. As mentioned earlier, the HDI focuses on the ability and sustainability of human. But no matter the poor, the rich, and even the developing or the developed countries, they must act under the constraints of the earth environment. Human actions and activities are carried out on the earth, and the impact of the actions of each country on its own country is subject to the natural conditions of the world. So, Bravo [17] considers that the environment

is also an important part of human sustainable development, and builds the HSDI by adding an indicator (per capita $CO_2$ emissions) to present environmental dimension based on the HDI, as is shown in Equation (7) and Table 6. Besides, with the process of human development, resource crisis has been exposed, especially the problems of excessive energy consumption and land pollution [18]. Thus, the ability and sustainability of human is under the constraints of the resource on the earth. From these considerations, the HGDI is constructed by adding some indicators both in resource and environmental dimensions (see Table 6). As we have defined, sustainable development is to coordinate the economic, social, and environment development, to balance the intra-generational welfare and maximize the total welfare of generations. Therefore, we should pursue economic growth to ensure the welfare of present generations, while protecting the ecological environment and rationally utilizing the natural resource to ensure the welfare of future generations. If we just want to protect the environment and make the economy stagnate, it is also not a sustainable development mode. Finally, the NSDI is built with economic, social and resource-environmental dimensions and 12 indicators (see Table 6).

$$HSDI = \sqrt[4]{I_{life} * I_{educaiton} * I_{income} * I_{emissions}} \tag{7}$$

**Table 6.** The relation and difference of indices.

| Index | Indicators | | | | Weight |
|---|---|---|---|---|---|
| | Economic | Environmental | Social | Resource | |
| HDI | Income | | Education<br>Life expectancy | | equal |
| HSDI | Income | $CO_2$ emissions | Education<br>Life expectancy | | equal |
| HGDI | Income | $CO_2$ emissions<br>PM10<br>Forest area (%)<br>Proportion of threatened animals (%)<br>Land conservation area (%) | Education<br>Life expectancy<br>Population using improved drinking-water sources (%)<br>Population using improved sanitation facilities (%)<br>Population below the minimum food energy (%) | Utilization ratio of primary energy (%) | equal |
| NSDI | Income<br>Economic growth<br>Economic structure | $CO_2$ emissions<br>PM2.5<br>Forest area (%) | Education<br>Life expectancy<br>population using improved drinking-water sources (%)<br>population using improved sanitation facilities (%) | Renewable energy consumption (%)<br>Arable land | Entropy Method |

We should note that although the NSDI has expanded the theoretical constraints of HDI, it has not denied HDI's theoretical tenet of human development, namely focusing on the ability and sustainability of human. On the contrary, the NSDI inherits the concept and indicators of HDI. For example, it includes the three indicators of the HDI. Additionally, the NSDI uses the Entropy method to calculate the weights of indicators, it is helpful to make up for the lack of objectivity that the equal weighted method used by HDI, HGDI and HSDI.

Figure 1 shows the correlations and corresponding scatter plots between the NSDI and other indices. The NSDI has positively correlation with HDI, but the correlation is weak (Pearson r = 0.398). It is mainly due to its structural composition, which is clearly more complex than that of the HDI, in which economic, social and environmental dimensions are considered. The NSDI has a stronger correlation with HGDI and HSDI than that with HDI. It indicates that HSDI and HGDI add indicators of environmental dimension to HDI, such as per capita $CO_2$ emissions, which is more systematic and comprehensive. In addition, NSDI is more comprehensive than HSDI and HGDI due to its structural composition with 3 dimensions and 12 indicators. Briefly speaking, HSDI or HGDI is an improvement of HDI, while NSDI is a further improvement based on those existing indices.

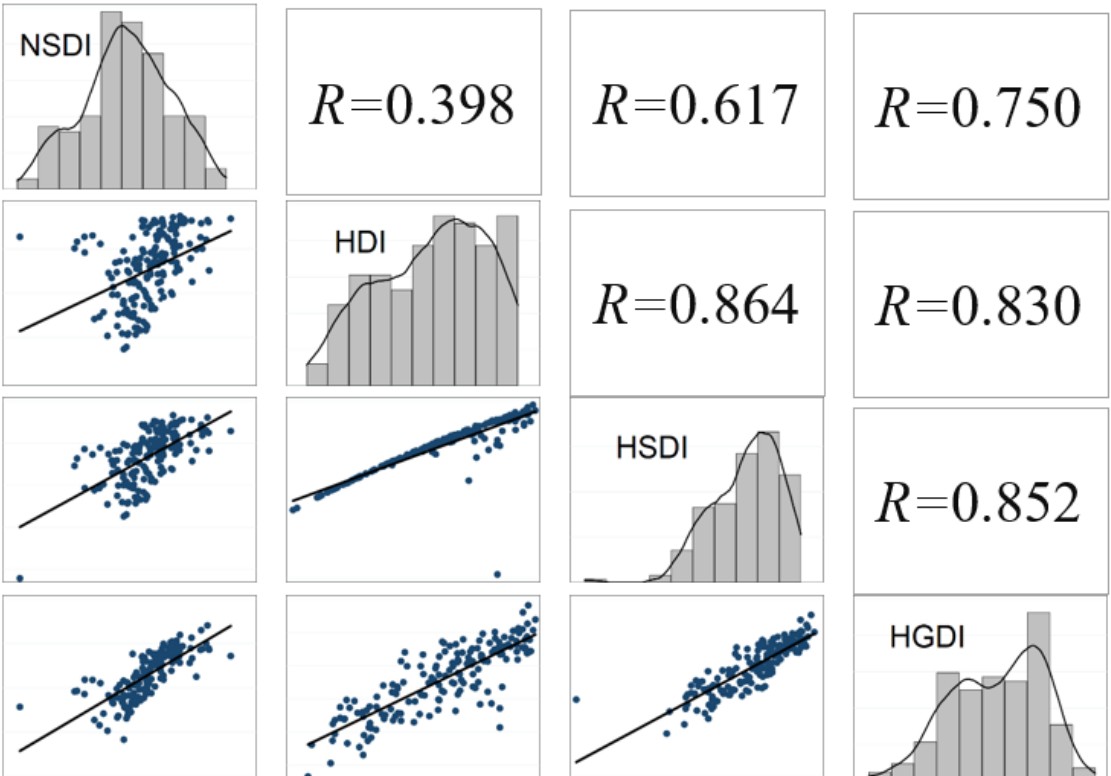

**Figure 1.** Binary correlations between the National Sustainable Development Index and other indices with corresponding scatterplots.

The HDI is correlated with HSDI and HGDI positively and strongly. Although the HSDI has added environmental indicator, it is still very close to the HDI, probably because the per capita $CO_2$ emission is not well enough to justified to represent the environmental dimension. As Estoque and Murayama [13] noted, whether the HSDI is fair enough to measure the complex human-environmental system remains to be questioned. On the contrary, the scatter diagram of HGDI and HDI shows that there is a big difference between HGDI and HDI despite of the strong correlation of them. This is mainly because HGDI has added 6 indicators from different aspects, which can better represent the sustainable development in resource and environmental dimension.

In addition, there is a strong and positive correlation between the HSDI and the HGDI. HSDI and HGDI, as two improvement schemes of HDI, both add environmental indicators to the HDI. Therefore, these two indicators have strong correlation is reasonable and in line with expectations.

Figure 2 shows the geographical distribution of the NSDI, HDI, HSDI and HGDI. It should be noted that the darker the blue, the higher the NSDI, HDI, HSDI or HGDI, while the white indicates the data vacancy. In general, the blue color of the countries in Western Europe, northern Europe, North America and Oceania is darker than that in Africa and Southeast Asia, which shows the blue color of the northern hemisphere is deeper than that of the southern hemisphere. It reflects the geographical distribution of the sustainable development level of the countries in the world, which means that the sustainable development level of the countries in Western Europe, northern Europe and North America is generally higher than that in Africa and Southeast Asia.

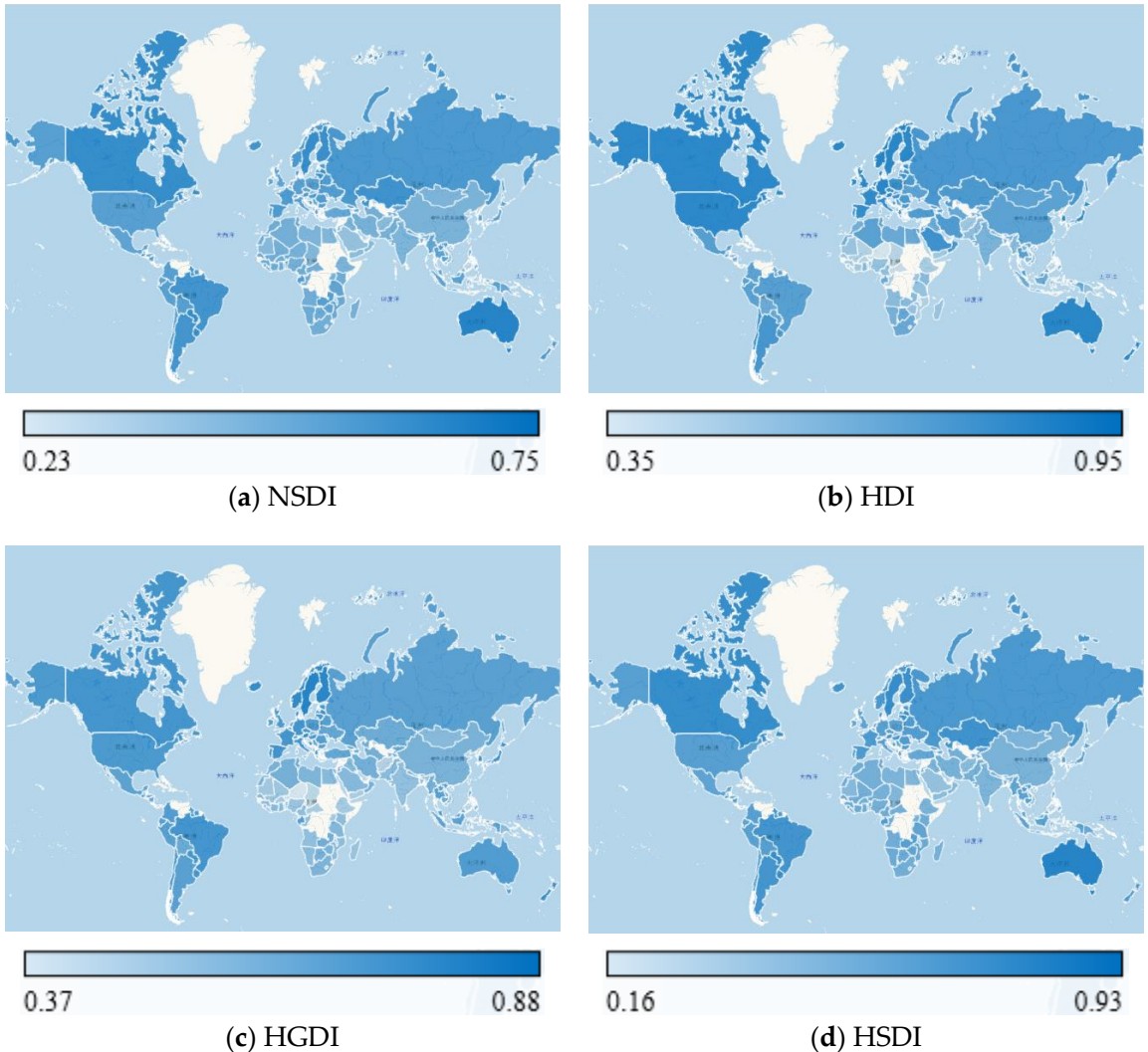

**Figure 2.** Geographical distribution of the 4 indices. Note: Subfigure (**a**) is the geographical distribution of the NSDI; Subfigure (**b**) is the geographical distribution of the HDI; Subfigure (**c**) is the geographical distribution of the HGDI; Subfigure (**d**) is the geographical distribution of the HSDI.

The rankings of some countries change quite drastically among NSDI, HSDI and HGDI. For example, the rankings of Australia and Canada in HSDI and HGDI are both lower than those in NSDI. The main reason is that, the $CO_2$ emissions per capita of Australia and Canada are more than 15 t, higher than most sample countries. While the HSDI and HGDI attach great importance to this indicator. But Australia and Canada perform very well in other indicators which plays significant role in NSDI, like education, health and so on. This makes the two countries have a high ranking in NSDI. Actually, this situation appeared in many counties, like those countries in the Middle East have been mentioned as follows.

The geographical distribution is similar in the 4 graphs, except for the Middle East. The Middle East countries are dark blue in Figure 2b and light blue in Figure 2a,c,d, like Qatar and Saudi Arabia. It shows that the Middle East countries have a high HDI ranking and low NSDI, HSDI and HGDI ranking. For example, Saudi Arabia ranks 34 in HDI, while NSDI, HSDI and HGDI rank 156, 75 and 124 respectively (see Table 7). This is mainly because the HDI does not include environmental indicators, while HSDI, HGDI and NSDI do. It is thus clear that the NSDI, HSDI, HGDI put a stop to the "celebration" of "gas-guzzling developed countries" [50].

**Table 7.** The comparison of NSDI and other index Rankings in 2015.

| Country | NSDI | HDI | HSDI | HGDI | Country | NSDI | HDI | HSDI | HGDI |
|---|---|---|---|---|---|---|---|---|---|
| Australia | 1 | 3 | 40 | 27 | Armenia | 83 | 71 | 56 | 79 |
| Norway | 2 | 1 | 3 | 5 | Kyrgyzstan | 84 | 103 | 95 | 90 |
| Switzerland | 3 | 2 | 7 | 1 | Sao Tome and Principe | 85 | 123 | 120 | 106 |
| Denmark | 4 | 9 | 12 | 3 | El Salvador | 86 | 101 | 94 | 84 |
| Canada | 5 | 11 | 17 | 34 | Gambia | 87 | 150 | 149 | 123 |
| Sweden | 6 | 5 | 1 | 2 | Vanuatu | 88 | 116 | 114 | 88 |
| Latvia | 7 | 39 | 9 | 25 | Papua New Guinea | 89 | 132 | 130 | 133 |
| Japan | 8 | 17 | 10 | 20 | Congo | 90 | 113 | 111 | 117 |
| United States | 9 | 12 | 31 | 37 | Malawi | 91 | 149 | 148 | 115 |
| Germany | 10 | 4 | 18 | 8 | Barbados | 92 | 55 | 42 | 76 |
| Serbia | 11 | 62 | 70 | 52 | Azerbaijan | 93 | 66 | 58 | 91 |
| Italy | 12 | 26 | 26 | 19 | Tunisia | 94 | 83 | 74 | 89 |
| Finland | 13 | 14 | 2 | 15 | Timor-Leste | 95 | 111 | 108 | 116 |
| New Zealand | 14 | 15 | 6 | 10 | Botswana | 96 | 88 | 84 | 104 |
| Lithuania | 15 | 35 | 22 | 24 | Samoa | 97 | 89 | 79 | 23 |
| France | 16 | 21 | 20 | 9 | Namibia | 98 | 110 | 106 | 134 |
| United Kingdom | 17 | 13 | 35 | 7 | Dominican Republic | 99 | 82 | 67 | 68 |
| Kazakhstan | 18 | 54 | 87 | 76 | Iran | 100 | 57 | 57 | 99 |
| Luxembourg | 19 | 20 | 42 | 45 | Maldives | 101 | 86 | 83 | 83 |
| Ireland | 20 | 6 | 39 | 6 | Ghana | 102 | 120 | 117 | 131 |
| Belgium | 21 | 16 | 32 | 12 | Suriname | 103 | 84 | 77 | 28 |
| Portugal | 22 | 38 | 15 | 28 | Lebanon | 104 | 69 | 65 | 85 |
| Iceland | 23 | 8 | 4 | 4 | Morocco | 105 | 104 | 98 | 110 |
| Netherlands | 24 | 10 | 48 | 13 | Cameroon | 106 | 130 | 128 | 140 |
| Korea (Rep.) | 25 | 22 | 27 | 29 | Tajikistan | 107 | 109 | 103 | 107 |
| Argentina | 26 | 44 | 66 | 36 | Jordan | 108 | 80 | 68 | 92 |
| Malta | 27 | 27 | 59 | 21 | Rwanda | 109 | 139 | 136 | 120 |
| Spain | 28 | 24 | 14 | 16 | Haiti | 110 | 144 | 143 | 138 |
| Israel | 29 | 19 | 61 | 18 | Senegal | 111 | 146 | 145 | 129 |
| Singapore | 30 | 7 | 46 | 14 | Kenya | 112 | 124 | 121 | 137 |
| Brazil | 31 | 67 | 16 | 55 | Peru | 113 | 72 | 60 | 62 |
| Belize | 32 | 87 | 41 | 78 | Angola | 114 | 125 | 125 | 144 |
| Montenegro | 33 | 47 | 11 | 39 | Togo | 115 | 143 | 141 | 154 |
| Fiji | 34 | 78 | 25 | 62 | Eswatini | 116 | 121 | 119 | 100 |
| Austria | 35 | 18 | 5 | 11 | Benin | 117 | 140 | 138 | 150 |
| Estonia | 36 | 28 | 13 | 49 | Cabo Verde | 118 | 107 | 101 | 109 |
| Greece | 37 | 29 | 29 | 23 | South Africa | 119 | 96 | 102 | 112 |
| Belarus | 38 | 50 | 56 | 47 | Comoros | 120 | 142 | 140 | 127 |
| Gabon | 39 | 92 | 55 | 88 | Mali | 121 | 157 | 157 | 153 |
| Hungary | 40 | 41 | 45 | 32 | Burkina Faso | 122 | 160 | 159 | 156 |
| Brunei Darussalam | 41 | 36 | 34 | 87 | Turkmenistan | 123 | 91 | 107 | 102 |
| Romania | 42 | 51 | 53 | 40 | Nigeria | 124 | 135 | 133 | 152 |
| Bulgaria | 43 | 48 | 58 | 41 | Oman | 125 | 45 | 69 | 103 |
| Lao PDR | 44 | 117 | 65 | 115 | Madagascar | 126 | 136 | 134 | 151 |
| Croatia | 45 | 43 | 24 | 35 | Lesotho | 127 | 138 | 137 | 147 |
| Ukraine | 46 | 73 | 78 | 72 | Nepal | 128 | 128 | 126 | 125 |
| Bhutan | 47 | 115 | 44 | 113 | Pakistan | 129 | 129 | 127 | 135 |
| Slovenia | 48 | 23 | 8 | 17 | Moldova | 130 | 95 | 86 | 94 |
| Algeria | 49 | 70 | 101 | 63 | China | 131 | 75 | 80 | 108 |
| Russian | 50 | 46 | 64 | 59 | Uganda | 132 | 141 | 139 | 157 |
| Slovakia | 51 | 37 | 33 | 30 | Bangladesh | 133 | 119 | 116 | 132 |
| Albania | 52 | 63 | 43 | 46 | Libya | 134 | 93 | 99 | 114 |
| Bosnia and Herzegovina | 53 | 68 | 47 | 66 | Equatorial Guinea | 135 | 118 | 122 | 145 |
| Turkey | 54 | 58 | 81 | 51 | Iraq | 136 | 102 | 100 | 136 |
| Bolivia | 55 | 100 | 86 | 93 | Solomon Islands | 137 | 131 | 129 | 95 |
| Colombia | 56 | 76 | 49 | 61 | Guyana | 138 | 106 | 105 | 52 |
| Uruguay | 57 | 53 | 19 | 38 | Egypt | 139 | 97 | 89 | 126 |
| Honduras | 58 | 114 | 73 | 112 | Mauritania | 140 | 137 | 135 | 155 |
| Cambodia | 59 | 126 | 93 | 123 | Qatar | 141 | 33 | 163 | 149 |
| Georgia | 60 | 64 | 57 | 50 | Guinea-Bissau | 142 | 152 | 151 | 113 |
| Czechia | 61 | 25 | 30 | 26 | Afghanistan | 143 | 145 | 142 | 161 |
| Poland | 62 | 32 | 51 | 33 | Guinea | 144 | 154 | 153 | 142 |

**Table 7.** *Cont.*

| Country | NSDI | HDI | HSDI | HGDI | Country | NSDI | HDI | HSDI | HGDI |
|---|---|---|---|---|---|---|---|---|---|
| Guatemala | 63 | 108 | 72 | 104 | Sierra Leone | 145 | 159 | 158 | 148 |
| Indonesia | 64 | 98 | 71 | 90 | Yemen | 146 | 147 | 146 | 160 |
| Panama | 65 | 61 | 37 | 44 | Congo (Dem. Rep.) | 147 | 153 | 152 | 141 |
| Chile | 66 | 40 | 38 | 31 | Zimbabwe | 148 | 133 | 132 | 111 |
| Mongolia | 67 | 79 | 128 | 82 | United Arab Emirates | 149 | 31 | 91 | 98 |
| Cyprus | 68 | 30 | 50 | 22 | Chad | 150 | 161 | 160 | 162 |
| Paraguay | 69 | 90 | 21 | 81 | Ethiopia | 151 | 151 | 150 | 158 |
| Bahamas | 70 | 49 | 54 | 43 | Liberia | 152 | 155 | 154 | 122 |
| Ecuador | 71 | 74 | 60 | 64 | Central African Republic | 153 | 162 | 161 | 159 |
| Mexico | 72 | 65 | 69 | 54 | Burundi | 154 | 158 | 156 | 143 |
| Malaysia | 73 | 56 | 36 | 53 | Trinidad and Tobago | 155 | 59 | 144 | 97 |
| India | 74 | 112 | 146 | 109 | Saudi Arabia | 156 | 34 | 75 | 124 |
| Thailand | 75 | 77 | 67 | 70 | Zambia | 157 | 122 | 118 | 96 |
| Jamaica | 76 | 81 | 74 | 71 | Nicaragua | 158 | 105 | 97 | 80 |
| Tanzania | 77 | 134 | 121 | 131 | Bahrain | 159 | 42 | 96 | 118 |
| Mauritius | 78 | 60 | 63 | 48 | Kuwait | 160 | 52 | 110 | 119 |
| Viet Nam | 79 | 99 | 75 | 92 | Mozambique | 161 | 156 | 155 | 139 |
| Philippines | 80 | 94 | 82 | 85 | Niger | 162 | 163 | 162 | 163 |
| Tonga | 81 | 85 | 77 | 73 | Cote d'Ivoire | 163 | 148 | 147 | 130 |
| Myanmar | 82 | 127 | 105 | 124 | | | | | |

From the analysis above, NSDI, HSDI, and HGDI are all modifications or improvements of HDI. The HSDI adds per capita $CO_2$ emissions to HDI, which is a breakthrough of HDI in the environmental dimension. The HGDI has a number of resource and environmental indicators, which can not only reflect sustainable development in the environmental dimension, but also represent the sustainable utilization of resources, while the NSDI fully considers the dimensions of economy, society, and resources and environment, which helps to measure the sustainable development level of a country from a more comprehensive perspective. To sum up, the NSDI represents a small step ahead from the HDI, HSDI, and HGDI.

## 6. Conclusions

This paper defines that sustainable development is to coordinate economic, social, and environment development, to balance intra-generational welfare, and to maximize the total welfare of generations. Thus, according to this concept, this paper built the National Sustainable Development Index (NSDI), involving dimensions of economy, society, and environment as well as 12 relative indicators, as an improvement of HDI. From the measurement of NSDI and the comparison of NSDI with other indices, we found that NSDI is a reliable and relative complete index for sustainable development assessment.

The NSDI makes up for the shortcomings of existing indices in some aspects. Firstly, HDI and HGDI equally weight all indicators, while the NSDI calculates the weight of each indicator by the entropy method, which helps to provide a battery of more objective weights. Secondly, HSDI and HGDI amend HDI by adding environmental indicators, but they are still not complete enough. According to the concept of sustainable development, we should coordinate welfare between the present and future generations, neither damaging the welfare of future generations nor giving up the reasonable utilization of resources for the sake of the welfare of future generations, and thus losing the welfare of the present generations. Therefore, a complete and objective index of sustainable development should take the three dimensions of economy, society, and resources and environment into account, like the NSDI does.

Based on the results above, we derived the following policy implications. Governments should be committed to promoting coordinated development in the dimension of economy, society, and environment, without "care for this and lose that". Specifically, governments should accommodate the business climate to help economic growth, then strengthen environmental protection and

resource utilization supervision, and finally improve the government spending on livelihood projects, especially education, medical care, and social security.

However, there are some limitations to this paper. Actually, a number of researchers have utilized the entropy method to weigh the indicators of a composite index, like Ma et al. [29], Wang et al. [46], and so on, but no one has compared the equal-weighted method with the entropy method. Honestly, it is difficult to compare the old method with the new method, and hard to say which one is better. Hence, we have to conduct a new study to compare the two methods exclusively, which will be presented in another forthcoming article.

**Author Contributions:** This paper was written by H.J. in collaboration with all co-authors. Data was collected by H.J. The first and final drafts were written by H.J., T.C., and X.Q. The results were analyzed by H.Z. and H.J. The research and key elements of the models were reviewed by H.J. and T.C. The writing work for corresponding parts and the major revisions of this paper were completed by H.Z. and T.C. All authors have read and agreed to the published version of the manuscript.

**Funding:** The authors acknowledge funding support from College Students' Science and Technology Innovation Activity Plan and Xinmiao Talent Plan Project of Zhejiang Province (No. 2018R43071).

**Acknowledgments:** The authors would like to express sincere gratitude to them, Giangiacomo Bravo from Linnaeus University, Vladimir Strezov from Macquarie University, and the peer reviewers, for their valuable suggestions; as well as Nikola Milic and other editors for their warm encouragement and meticulous work.

**Conflicts of Interest:** The authors declare no conflict of interest.

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
