# Peer review of "A Global Assessment of Sustainable Development Based on Modification of the Human Development Index via the Entropy Method"

_sustainability, doi:10.3390/su12083251_

Round 1
Reviewer 1 Report
Referee Report on “A Global Assessment of Sustainable Development: Based on the modification of Human Development Index with Entropy method _726120
Thanks to the author for the correction and explanation. However, there are several suggestions.
Major Concerns and Comments:
- As in the previous review, the author is arbitrary in the selection of indicators. For example, why was the Proportion of threatened animals, Land conservation area and Population below the minimum food energy deleted? Why should Utilization ratio of primary energy be replaced by Renewable energy consumption and? Why should economic growth and economic structure be added? I think the author should explain carefully.
- The ranking of some countries (such as Australia and Canada) has changed quite drastically among NSDI, HSDI and HGDI. The author should carefully explain the implications of these dramatic changes. The reason for the ranking change is due to the selection of indicators or the adjustment of weights, and the author should explain.
- I think that the preparation of NSDI and the calculation of weights are quite meaningful. The author should highlight the meaning of the index itself and explain the calculated results. I suggest moving Appendix 1 (The weights of 12 indicators) to Section 4 for discussion, which is an important result of this research.

Reviewer 2 Report
The article has been reworked according to my comments in the review report and I am equally satisfied with the authors' comments on my specific comments.
Author Response
Thank you for your helpful and pertinent comments again.
Reviewer 3 Report
As far as I see, you have responded adequately to my recommendfations.
Author Response

(The authors gave the same response as above.)

Reviewer 4 Report
Dear Authors,
Thank you for giving me the opportunity to serve as a reviewer. I learned a lot from your manuscripts and the response letter. I believe that the updated version is much readable and understandable. I wish you the very best in publishing this research paper.
Author Response

(The authors gave the same response as above.)

Reviewer 5 Report
In the resubmitted version of the paper, like in the original one, the Authors propose an aggregate index of sustainable development, based on modifications of the Human Development Index. The Authors choose diagnostic variables for the three pilars of sustainable development (social, economic and environmental) to be included into the composite index. They have calculated the National Sustainable Development Index according to proposed methodology and compared performance of 163 countries. Their work aims at addressing existing challenges in measuring sustainable development.
The re-submitted version includes some changes comparing to the previous one, however the paper still should be improved.
Detailed comments:
Introduction - P.2, par. 2, lines 1-4 (a comment repeated from the first version): The Authors state that HDI has several drawbacks as a SD index: The authors of HDI haven't designed it as a sustainable development index and therefore one cannot "blaim" HDI that it does not measure sustainability. Of course, this does not diminish the fact that measuring sustainability is a huge challenge.
The Authors' starting point is HDI, but they want to offer something substantially different - an aggregate index of sustainable development.
The review of indicators used for measuring development is not clear - monetary indicators (including ISEW) are not distinguished from the composite ones (e.g HDI) or expressed in other units (e.g. Ecological Footprint). Apart from the HDI, the Authors refer to two aggregate indexes related to sustainable development (HGDI and HSDI,), but missed at least one important aggregate sustainable development indicator: Sustainable Development Index published by the United Nations. https://www.sustainabledevelopment.report/ (the remark repeated from the original version)
In my opnion, the comparison of composite indexes of sustainable development done as a part of Discussion (pages 11-12), could be discussed in the Introduction as a starting point for the design of the index proposed by the Authors. What are the gaps in existing indicators? Design of the proposed NSDI should address these gaps.
In the literature review the Authors discuss the concept of sustainable development (chapter 2) refering mainly to the classic definition as formulated in Bruntland Report. They conclude that "sustainable development is to coordinate the economic, social, and environment development, to balance the intra-generational welfare and maximize the total welfare of generations." What is needed here is the link between the concept of sustainable development and an indicator they propose. The Authors cover three pilars of SD by diagnostic variables, but don't discuss other cirteria in detail. What is the consequence of the principle of intergenerational equity for the construction of the index? In what way the teoretical criteria of environmental sustainability are addressed by the proposed variables?
In my opinion, the literature review could pay more attention to the problems related to measuring sustainable development rather than the concept itself.
The construction of the NSDI is quite transparent, however the justification for choosing particular diagnostic variables (sub-chapter 3.2) is rather general. Only one field - education - was discussed in detail in the sub-chapter 3.3. In my opinion, the choice of other variables should be described in a similar way. In particular, conclusions from the critique of existing indexes and sustainability criteria discussed in the Chapter 2 should be considered.
In the chapter 4. The measurement of NSDI (paragraph 2), the Authors say that "we should pursue economic growth". Such a statement is to a large extent controversial - the continuing economic growth (measured as increase in production and consumption), especially in highly developed countries, is perceived as a reason for the current unsustainable development path. For this reason, in HDI the fixed upper limit of income per capita (75.000 USD) is taken. In Table 2 the Authors don't explain their choice and meaning of GDP growth as a measure of economic sustainability.
Additionaly, when the Authors discuss the construction of the NSDI, they don't mention a problem of correlation between variables (the comment repeated from the previous review).
For the normalization (page 7), the Authors apply simple min-max normalization without considering the recent developments e.g. in HDI methodology, namely using fix minima and maxima where possible, especially fix maximum income per capita (comment repeated).
The results have been presented in a transparent manner.
Disucssion
I appreciate that the Authors have added the appendix with weights obtained for variables. They discuss the weight of three pillars of SD. In my opinion, a very important result is that the sum of weights of the economic and social dimension is almost equal to the weights of the resource-environmental dimensions. In the Authors' opinion this confirms theoretical consideration. Perhaps it is also worth mentioning that environmental resources are an important factor of economic development and contributes to quality of life which justifies this high weight.
Author Response
Please see the attachment.

This manuscript is a resubmission of an earlier submission. The following is a list of the peer review reports and author responses from that submission.
Round 1
Reviewer 1 Report
Referee Report on “A Global Assessment of Sustainable Development: Based on the modification of Human Development Index with Entropy method _726120
This study constructs the National Sustainable Development Index (NSDI), explores the degree of international sustainable development, and uses the entropy method to deal with the weighting of indicators. First, HDI and HGDI have equal weights for all indicators, and NSDI calculates the weight of each indicator through the entropy method, which helps to obtain more objective weights. Secondly, NSDI integrates three indicators of HDI, HSDI and HGDI, including economic, resource, environmental and social dimension, which can better reflect ecological, economic and social development.
I think this is a meaningful paper for measuring sustainable development. However, there are several suggestions.
Major Concerns and Comments:
- As the author said, this paper has two main contributions: First, it combines the advantages of the three indexes to comprehensively evaluate ecological, economic, and social development; second, it uses the entropy method to deal with the weighting of indicators. However, due to the comprehensive adjustment, it is difficult to compare the old method with the new method. Is the ranking of the comprehensive index among countries changed due to the selection of indicators or weighted weights?
- The authors combined the indicators of the three indexes of HDI, HSDI and HGDI. In addition to deleting some indicators, they also added different indicators. As the author points out, the indicators chosen should be representative, objective, concise and acceptable. However, I don't know whether the author has analyzed the relationship between indicators when deleting or adding some indicators. Why does the author not allow the indicators of HDI, HSDI, HGDI plus the indicators added by the author to calculate the weight together, and then delete unnecessary indicators?
- The author uses three aspects of economy, environment and society to construct NSDI indicators, and then measures the comprehensive sustainable development of each country. However, the authors lacked discussion about the relevance of the three dimensions, which could not lead to policy implications. Authors are suggested to discuss moderately to enrich the contribution of this research.
Minor Concerns
The sixth line of the fifth paragraph on the third page, “…Kondyli (2010), and Liu et al. (2015), Liu et al. (2017).” should be “…Kondyli (2010), Liu et al. (2015), and Liu et al. (2017).”
Evaluation:
For the above reasons, I believe that this article can be published with appropriate modifications.

Reviewer 2 Report
The article deals with a very interesting and significant area of ​​global comparison evaluation by adjusting the Human Development Index (HDI).
The paper is processed at a very good scientific research level, but I miss a more detailed description of the differences in the indicators between the applied indices such as HDI, HSDI, HGDI within a single comparative table.
In the article, it would be appropriate to add specific differences in the indicators applied in the proposed NSDI index to those applied in the research.
Although the new NSDI has three dimensions, the number of indicators in each dimension is not the same (dimension I have 3 indicators, dimension II has 5 indicators, and dimension 3 has 4 indicators), which can lead to a different ranking of individual dimensions and ultimately impact on overall rating using the NSDI index.
In the article, I have noticed a mislabeling of the name of Table 1 with respect to the text published above, which states that it is a description of the input indicators for the new NDSI.
I recommend reviewing bibliographic links with a cross-link to the current bibliographic list as instructed by the authors of a particular journal.
Reviewer 3 Report
You must review and correct the paper for mistakes in English usage (many present, involving misuse of verbs, nouns, adjectives). "Entropy method" uses an "entropy idea of entropy": what does that mean? This method seems a key feature of your article, but, if so, you must explain it clearly, and avoid truism in doing so.
Reviewer 4 Report
Journal: Sustainability (ISSN 2071-1050) Manuscript ID: sustainability-726120 Title: A Global Assessment of Sustainable Development: Based on the modification of Human Development Index with Entropy method Review Report Using the Entropy method, this study compares four development indices - Human Development Index (HDI), Human Sustainable Development Index (HSDI), Human Green Development Index (HGDI), and National Sustainable Development Index (NSDI). It measures the NSDI of 163 countries and validate the effectiveness of the proposed index compared to that of other existing indices. The documented results show that the NSDI is a more reliable and comprehensive index for sustainable development assessment, which has made up for the shortcoming of existing indices. In sum, I really enjoy reading the manuscript and learning from this research project about various development indices. I believe that it provides a new opportunity to think about how we can improve the quality of practical indices. My only remaining question is the following: Why the corresponding scatterplots between NSDI and HSDI and between HSDI and HGDI are same on the reported Figure 1 on page 10? It could be a typo but please fix the Figure 1. Again, I thank you for giving me the opportunity to read and learn from this project. Regards,Author Response
Please see the attachment.

Reviewer 5 Report
The goal of the paper is to propose an aggregate index of sustainable development, based on modifications of the Human Development Index. The Authors made an attempt to identify sustainability criteria and corresponding diagnostic variables that should be included into the composite index. They have proposed the National Sustainable Development Index. Their work tries to address existing challenges in measuring sustainable development, however does not meet quality criteria.
Detailed comments:
The literature review has a broad scope, but seems not to adress problems with measuring sustainable development properly.The discussion concerning the concept of sustainable development (chapter 2) is poorly linked to the goal of the paper, i.e. aggregate index of sustainable development. In my opinion, there should be more attention paid to the problems related to measuring sustainable development rather than the concept itself.
The Authors refer to two aggregate indexes related to sustainable development (HGDI and HSDI, p.2, last paragraph), but missed at least one important aggregate sustainable development indicator: Sustainable Development Index published by the United Nations.
P.2, p 2, lines 1-4: The Authors state that HDI has several drawbacks as a SD index: I dare to claim, that HDI is not a sustainable development index and therefore one cannot "blaim" HDI that it does not measure sustainability. The Authors' starting point is HDI, but they want to offer something substantially different - an aggregate index of sustainable development.
The construction of the NSDI is quite transparent, however the justification for choosing particular diagnostic variables (sub-chapter 3.2) is rather general. The choice seems to be rather arbitrary. In particular, there is little reference to the critique of existing indexes which was done in the introduction and sustainability criteria discussed in the Chapter 2. For example, how the proposed economic variables address the Authors' claim, that "for the economic dimension of sustainable development, the governments should pursue a relatively high and fairness income for folks"?
How is the need for intergenerational fairness addressed?
It has been stated (page 10, last chapter), that "per capita CO2 emission is not well justified to represent the environmental dimension" and later this variable is included in their NSDI.
Additionaly, when the Authors discuss the construction of the NSDI, there is a problem of correlation between variables. Have the Authors analysed this?
For the normalization (page 7), the Authors apply simple min-max normalization without considering the recent developments e.g. in HDI methodology, namely using fix minima and maxima where possible, especially fix maximum income per capita.
The results have been presented in a transparent manner (although the Figure 2 is not legible).
Conclusions
The Authors say that" the composite indicators are comprehensive and representative, and each indicator constitutes an organic related system", but they really haven't confirmed that. What criteria for the representativeness and comprehensiveness have they applied?
Very interesting result are the weights obtained for variables, but Authors haven't discussed them at all. It would be worth explaining what are the weights of the three dimensions of sustainable development and what it actually means in the context of the concept itself. What is the reason for e.g. such a high weight ot the arable land per capita?